# Capacity Planning of Timber Harvesting in Windthrow Areas

**Martin Kühmaier** [1,*] , **Christoph Gollob** [2] , **Arne Nothdurft** [2] , **Maximilian Lackner** [3] and **Karl Stampfer** [1]

[1] Institute of Forest Engineering, University of Natural Resources and Life Sciences, Vienna, Peter-Jordan-Str. 82, 1190 Vienna, Austria; karl.stampfer@boku.ac.at

[2] Institute of Forest Growth, University of Natural Resources and Life Sciences, Vienna, Peter-Jordan-Str. 82, 1190 Vienna, Austria; christoph.gollob@boku.ac.at (C.G.); arne.nothdurft@boku.ac.at (A.N.)

[3] Department of Industrial Engineering, University of Applied Sciences Technikum Wien, Höchstädtplatz 6, 1200 Vienna, Austria; maximilian.lackner@technikum-wien.at

* Correspondence: martin.kuehmaier@boku.ac.at; Tel.: +43-1-47654-91518

**Abstract:** Digitization can help the forest industry to improve cost efficiency and to reduce possible environmental impacts. In the context of this study, models were implemented using the example of windthrow processing, which enables a capacity planning for carrying out timber harvesting. For capacity planning, it is necessary to estimate the time required by the harvesting systems. For this purpose, existing productivity models were analyzed, the models were validated and adjusted, and the time required for each harvesting system and calamity area was calculated using stand and terrain parameters. Depending on the scenario and the preferred harvesting system, the time for harvesting the timber in an almost 200-hectare windthrow area in a case study region in Carinthia (Austria) varied. The harvesting with cable yarder and tractor takes about 26,000 machine hours and 86,000 man-hours. Harvesting operations with cable yarder and harvester-forwarder has proven to be the most productive with a duration of around 20,000 machine hours and 70,000 man-hours. Depending on the scenario, in windthrow areas, forest workers are needed for 28 to 42 min to fell, delimb, buck and extract 1 m$^3$ of timber to the forest landing.

**Keywords:** windstorm; windfall; post-disturbance; forest management; wood harvesting; productivity models; cable yarder; forwarder

## 1. Introduction

Due to climate change, extreme weather events such as storms are increasing. Storms and strong winds have caused major economic losses in forestry in Central and Northern Europe since the 1990s [1]. For example, in the period from 1950 to 2000, an average of 35 million m$^3$ of timber was exposed to various abiotic and biotic damage annually, which corresponds to 8.1% of the total annual logging in Europe and 0.15% of the total stock during this period [1]. Between 1950 and 2000, windthrows caused 53% of the total damage. It is expected that the frequency and severity of storms will continue to increase due to climate change in Europe [2].

Most wind damage occurs in mature stands in the immediate vicinity of recently harvested stands or in recently heavily thinned stands in which trees have not yet become accustomed to the increasing wind load [3]. Spruce (*Picea abies* Karst.) with shallow roots is more susceptible to wind damage than pine (*Pinus sylvestris* L.) [4]. The risk of wind damage can be reduced by avoiding sharp transitions between old stands and harvested areas, and by first using the older stands that are most at risk [3,5–8].

Secondary damage from forest pests and loss of quality regularly occur after large-scale windthrows. In order to keep this as low as possible and to compensate for the economic loss caused by the damaged trees, it is recommended to deal with the windthrows as quickly as possible. For the planning of the processing, it is essential to be aware of the amount of damaged wood and its spatial distribution, as it depends on how many workers

and machines are required for processing, of which quantities must or can be purchased by the saw and paper industry in which period and areas should be given priority [9]. Holistic logistic planning of the harvesting operations does not often take place, but there is strong competition for the existing capacities of timber harvesting and transport companies. For improved planning, decision support tools are needed that enable a spatially explicit estimate of the volumes of damaged wood and provide guidelines for efficient processing. This is particularly necessary for crisis management, where quick and efficient timber harvesting and logistics should take place.

Climate change, which is currently one of the greatest challenges in the forest industry, will increase the pressure for digitization even further. The most urgent task is to apply existing technologies and solutions in forest production systems and forest practice [10]. Predicting the amount of damaged wood is usually very difficult and based on subjective estimates. The integration of digital information, for example from ALS (airborne laser scanner) and TLS (terrestrial laser scanner) data, harbors considerable potential for improvement in the planning, execution and quality assurance of timber harvesting and logistics. The amount of damaged timber after windthrows could be determined in terms of quantity and on spatial level through comparative analyses of current and historical ALS data. This information would be suitable as a basis for a classification of the suitability of timber harvesting systems with subsequent logistical optimization [9]. A developed model could be used for future damage events that occur more and more frequently in order to obtain a more objective derivation of the volumes of damaged wood, and also to be able to technically and economically optimize harvesting operations in windthrow areas.

The objective of this paper is to predict which harvesting systems can be used on windthrow areas from a technical point of view, how many machines and workers are needed to execute harvesting operations on windthrow areas and how long the operations will last.

## 2. Predicting the Efficiency of Timber Harvesting Operations after Windthrows

Especially in the last ten years, there have been some studies carried out in order to better assess the efficiency and costs and support the planning of timber harvesting operations after windthrows [2,11–22]. In this section, already developed productivity models are presented. If not already included in the model, a correction factor of 0.7 was assumed for the conversion from productive system hour without breaks ($PSH_0$) to productive system hour with breaks lower than 15 min ($PSH_{15}$). The influencing variables were mostly stand, terrain and operational parameters (Table 1).

**Table 1.** Parameters used for the calculation of the productivity of timber harvesting operations after windthrows.

| Parameter |
| :---: |
| average tree volume in m$^3$ |
| average stand density in trees per hectare |
| average breast height diameter in cm |
| average load volume in m$^3$ |
| average extraction distance in m |
| average log volume in m$^3$ |
| maximum terrain slope while driving in % |
| slope of the machine during loading in % |
| maximum terrain slope during loading in % |
| number of logs per load |
| average lateral yarding distance in m |
| average slope of the terrain in degrees |
| degree of damage to the thrown trees (1 is uprooted, 2 is uprooted and broken) |

Kärhä et al. [2] analyzed the productivity and costs of windthrow processing with harvesters (Ponsse Ergo, John Deere 1270D ECOIII, Logset 8H GT) in Finland and compared



this data with the efficiency of timber harvest in undamaged stands. In both cases, stocking was done with the harvester without exception. The results showed that the productivity of windthrow operations was 19 to 33% lower than that of undamaged areas. On windthrow areas, the costs for processing with a harvester for trees with a volume of 0.3 to 1.5 $m^3$ were 35 to 64% higher. A productivity model for harvesters was developed for tree volumes of 0.3 to 1.5 $m^3$ (average: 0.7 $m^3$) and stand densities from 284 to 708 trees per hectare (average: 422 trees per hectare).

Brzózko et al. [12] examined the productivity of harvesters (Rottne H-14, Valmet 941, Valmet 911) during windthrow processing in Przedbórz (Poland). According to the authors, the stocking was done exclusively with the harvester. The productivity achieved was around 40–60% lower compared to conventional timber harvesting under similar natural conditions. This was due to the unique nature of the post-windthrow site, which included conditions that would not occur with conventional harvesting and reduced productivity (e.g., different types of tree damage requiring different operator approaches). The operator's experience and machine specification were additional factors that influenced productivity. A productivity model for harvesters was developed for an average tree volume of 0.32 $m^3$ and an average stand density of 406 trees per hectare.

Dvořák [13] evaluated the performance of a Ponsse Ergo-Harvester when processing windthrow in the Czech Republic. Here, too, the stocking was done exclusively with the harvester. The time required to process a tree was between 62 and 171 s, depending on the tree volume. The productivity was between 6.5 and 28.0 $m^3$ per $PSH_0$ with a trunk volume of 0.1 to 1.2 $m^3$. A productivity model was developed for tree volumes from 0.10 to 2.00 $m^3$ (average: 0.98 $m^3$). With a tree volume of 0.98 $m^3$, a productivity of 18.9 $m^3/PSH_{15}$ was achieved.

Hagauer [11] carried out a study of windthrow processing with harvesters in Switzerland. In this case, two to four experienced forest workers were stocking trees by chainsaw, followed by straightening, delimbing and cutting using an excavator and harvester. The tree diameter in breast height (DBH) was identified as the only significant influencing variable for reprocessing with a harvester. With an average DBH of 37 cm, a productivity of around 40 $m^3$ per hour was achieved. A productivity model was developed for tree volumes from 0.04 to 6.57 $m^3$ (average: 1.97 $m^3$) and DBH from 11 to 65 cm (average: 37 cm). With a tree volume of 1.97 $m^3$, a productivity of 37.39 $m^3/PSH_{15}$ was achieved.

Cadei et al. [19] carried out an evaluation of windthrow processing with forwarders on three areas in northeastern Italy. The time studies carried out comprised 59.9 h and 101 work cycles, with a total volume of 1277 $m^3$ of wood being harvested. The average productivity for the three locations was 22.5, 18.5 and 29.4 $m^3$ per $PSH_{15}$, respectively. The average log volume, the inclination of the machine during loading and the number of logs had a positive effect on productivity. On the other hand, the moving distance, the load volume, the slope of the terrain while driving and loading had a negative influence. With an average extraction distance of 500 m, a productivity of 16 to 23 $m^3$ per $PSH_{15}$ could be reached. An increase in the forwarding distance by 200 m leads to a higher time requirement and thus to a reduction in productivity of 6%. A productivity model was developed for load volumes from 1.9 to 24.9 $m^3$ (average: 12.93 $m^3$), hauling distances from 41 to 2221 m (average: 724 m), log volumes from 0.10 to 0.81 $m^3$ (average: 0.34 $m^3$), maximum slope during driving from 21 to 86% (average: 45%), slopes of the machine during loading from 20 to 96% (average: 54%), maximum slope during loading from 19 up to 95% (average: 39%) and 11 to 104 logs per load (average: 45 logs).

Iranparast Bodaghi et al. [20] examined the efficiency of timber harvesting with a skidder and tractor on two windthrow areas in Iran. The working team consists of three people including a choker setter, chainsaw operator, and skidder/tractor operator. The productivity of extracting with a skidder and tractor was relatively low at 1.54 and 0.81 $m^3$ per hour, respectively. Compared to conventional timber harvesting, the system productivity for windthrow processing was about 6 to 15 times lower. The harvesting costs for extracting with a skidder or tractor were around 60 or 100 euros per $m^3$, which was higher than with

conventional timber harvesting. In both cases, the harvesting costs were 10% to 30% above the selling price of the wood. The harvesting, therefore, has no direct economic justification, but can be important for other reasons like forest protection, new stand establishment, etc. A productivity model was developed for a skidder. The input data of the model is based on an average log volume of 1.90 m$^3$, load volume of 3.54 m$^3$, a lateral yarding distance of 45.8 m and a skid trail length of 308 m. Another productivity model was developed for a tractor. The input data of the model is based on an average log volume of 1.58 m$^3$, load volume of 2.54 m$^3$, an extraction distance of 49.8 m and a skid trail length of 195 m.

Borz et al. [15] developed two productivity models for the TAF 690 OP and TAF 657 skidders for windthrow areas in difficult terrain in Romania. Each skidder was served by two workers. One was felling the trees with chainsaw (when necessary) and partially processing the trees. Another one was doing landing operations, including final processing of the trees. The most significant influencing factors were the lateral yarding and extraction distance and the number of logs. With an average lateral yarding distance of 20 m and an extraction distance of 980 m, the time study showed a productivity of 3.75 m$^3$ h$^{-1}$ in the case of the TAF 690 OP. In comparison, in the case of TAF 657, with an average lateral yarding distance of 23 m and an average extraction distance of 871 m, the productivity was 3.20 m$^3$ h$^{-1}$. A productivity model was developed for the TAF 690 OP skidder for load volumes of 4.17 to 10.03 m$^3$ (average: 6.52 m$^3$), lateral yarding distances of 3.10 to 44.40 m (average: 19.90 m) and extraction distances of 128 to 1338 m (average: 980 m). Another productivity model was developed for the TAF 657 skidder for load volumes from 1.97 to 8.33 m$^3$ (average: 5.38 m$^3$), lateral yarding distances from 9.80 to 56.50 m (average: 22.86 m), extraction distances from 107 to 1526 m (average: 871 m) and 2 to 12 logs per load (average: 6.17 logs).

Stoilov et al. [22] developed a productivity model for a Koller K501 cable yarder mounted on a truck. The work team consisted of the yarder operator, a second worker who unhooked, delimbed, and bucked the trees and a choker-setter at the yarding site. The significant influencing factors were the extraction distance, the lateral yarding distance, the slope and the degree of damage to the trees that were thrown. With an average extraction distance of 101 m and a lateral yarding distance of 18 m, the time study showed a productivity of 20.1 m$^3$ h$^{-1}$. Taking breaks into account, productivity decreased to 12.8 m$^3$ h$^{-1}$. The following productivity model was developed for load volumes from 0.40 to 1.80 m$^3$ (average: 1.10 m$^3$).

## 3. Materials and Methods

### 3.1. Study Area

The case study was carried out in Carinthia (Austria). Five regions in the Hermagor district were selected in which the analysis of the research questions took place: Frohntal (46°39′50″ N 12°45′30″ E), Liesing (46°40′50″ N 12°48′50″ E), Laas (46°42′20″ N 12°58′40″ E), Mauthner Alm (46°38′50″ N 12°57′50″ E), Plöcken (46°37′00″ N 12°57′50″ E). There was a total of 564 harvesting spots with a total area of around 212 hectares. The study area was at an altitude of 868 to 1968 m above sea level and the mean slope was 69%. The forest road density in this region was about 70 m ha$^{-1}$. The mean extraction distance for forwarders was 217 m, and for cable yarders 154 m. On average, there were about 145 trees per hectare with an average tree volume of 2.21 m$^3$ and an average diameter at breast height of 32.9 cm. The average stand volume was about 544 m$^3$ per ha.

### 3.2. Data Set

The time required for harvesting operations had to be estimated for capacity planning. The time required was calculated by including productivity models (see Section 2). The extraction distance, terrain slope, tree volume, stand density, diameter of the mean basal area tree (proxy for DBH), lateral yarding distance, load volume and the number of logs per load served as input data for the used productivity models (Table 2).

**Table 2.** Descriptive statistics of the most important terrain and stand parameters.

| | S | ED1 | ED2 | DG | TV | V | SD |
|---|---|---|---|---|---|---|---|
| Average | 68.9 | 217 | 154 | 32.9 | 2.21 | 544 | 145 |
| Minimum | 4.3 | 5 | 4 | 0.0 | 0.00 | 11 | 0 |
| 5% Quantile | 25.4 | 19 | 14 | 19.4 | 0.49 | 218 | 5 |
| 25% Quantile | 48.9 | 47 | 34 | 28.1 | 1.27 | 394 | 14 |
| Median | 66.5 | 91 | 64 | 34.0 | 1.95 | 513 | 35 |
| 75% Quantile | 83.8 | 222 | 158 | 38.5 | 3.07 | 693 | 113 |
| 95% Quantile | 122.6 | 957 | 679 | 44.2 | 4.37 | 964 | 611 |
| Maximum | 178.7 | 1337 | 948 | 53.2 | 6.15 | 1123 | 3297 |
| Standard Deviation | 29.625 | 290.053 | 205.711 | 7.972 | 1.219 | 219.439 | 354.315 |

S = Slope (%), ED1 = Extraction distance for wheeled machines (m), ED2 = Extraction distance for cable yarder (m), DG = Diameter of the mean basal area tree (cm), TV = Tree volume ($m^3$), V = Wood volume ($m^3\ ha^{-1}$), SD = stand density (trees $ha^{-1}$).

A new method based on terrestrial and airborne laser scans within a Bayesian inferential framework for the rapid estimation of the amount of damaged timber after a windthrow event and its spatial distribution has been developed in Nothdurft et al. [23]. Next to the windthrow areas, undamaged forest stands were visited in May 2020, in which a total of 62 sample plots were measured with a person-carried laser scanner. With program routines developed at the Institute for Forest Growth at the University of Natural Resources and Life Sciences, the standing trees were measured fully automatically, and the standing wood volume was determined [24]. Aerial RGB images (ortho-mosaics in GeoTiff format) were obtained on behalf of the Carinthian forest service by several commercial operators on different platforms (UAV and airplane). On the basis of these orthophotos, a total of 564 windthrow areas (212.3 ha in total) were digitized.

For the 62 sample plots, as well as for the 564 windthrow areas, descriptive data was derived from a vegetation height model and from a terrain model, such as the diameter of the mean basal area tree, the wood volume per hectare or the mean vegetation heights. Regression models were then created for the wood volume on the sample areas as a function of these parameters. For this purpose, a new approach was developed with which spatial trends could be formulated using Bayesian techniques. The new estimation method was transferred to the windthrow areas in order to quantify the amounts of damaged wood. In addition, confidence levels were calculated. It turned out that, despite the relatively small sample size, very precise forecasts of the amount of damaged wood could be made. The relative mean squared error for most of the windthrow areas was below 20%. Finally, the estimated amounts of damaged wood and uncertainties were mapped (Figure 1). Likewise, to the forecast of the amount of damaged wood, forecasts were also made for the diameters of the mean basal area trees for the windthrow areas.

The tree volume is an important input parameter for the productivity models. The tree volume was calculated as follows:

$$TV = dg * MVH * 0.48 \tag{1}$$

TV represents the average tree volume in $m^3$, dg the diameter of the mean basal area tree in m, MVH the mean height of vegetation in m and 0.48 is a constant form factor. To calculate the stand density, the area in hectares was first calculated in [23]. Then the volume per hectare was multiplied by the area to get the absolute volume. In the final step, the absolute volume was divided by the average tree volume.

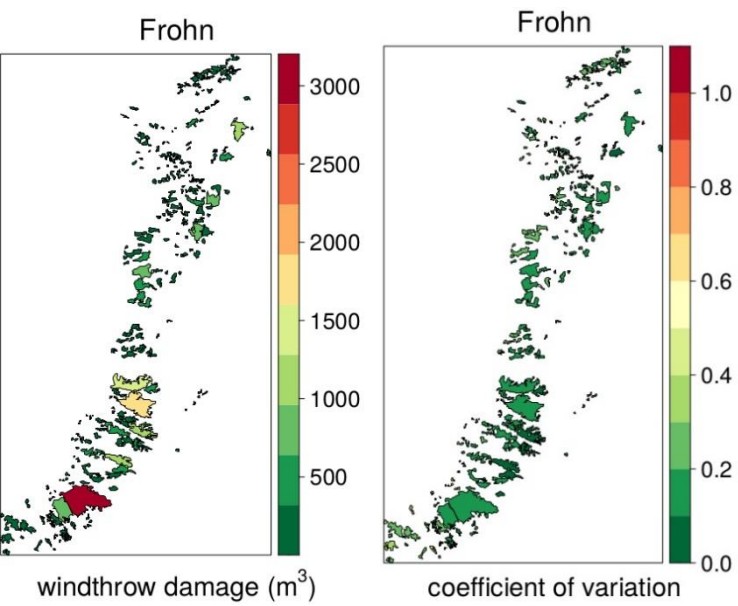

**Figure 1.** Forecast of the amount of damaged wood in m$^3$ (**left**) and uncertainty of the estimate (variation coefficient) (**right**) [23].

### 3.3. Capacity Planning

For capacity planning in windthrow harvesting, it is necessary to be able to estimate the time required by the individual timber harvesting systems. Productivity models for harvesters, forwarders, tractors, skidders and cable yarders are used to estimate the required working time for processing the trees and extracting them to the forest landing. The required working time is only calculated for those areas on which the respective system can be used technically based on a terrain classification, which is usually based on terrain parameters (slope, morphology, distance to the forest road) and the requirements of the harvesting systems (machine configuration, terrain requirements, etc.) [25,26]. The time required is calculated per m$^3$ and then multiplied by the total harvest volume of each calamity area. The sum of all areas gives the time required for the entire windthrow processing.

For capacity planning, the maximal annual utilization of the forest machines is relevant. The calculation is based on the empirical values in Table 3 [27]. The utilization of the cable yarder is relatively high but can be achieved with optimal planning. Harvesters and forwarders are sometimes used in two-shift operation and can then achieve even higher annual utilization rates than other machines; this would certainly be true in the case of extreme events (windthrow, calamities), where rapid processing is necessary.

**Table 3.** Annual utilization of forest machines.

| Machine Category | Hours |
| --- | --- |
| Harvester | 1600–2000 |
| Forwarder | 1400–1800 |
| Skidder | 800–1200 |
| Tractor | 500–800 |
| Cable Yarder | 1500 |

The harvesting system harvester-forwarder needs for every machine one and in total two operators, all other systems (chainsaw-skidder, chainsaw-tractor, chainsaw-cable yarder) need a team of three people. For the relocation of forest machines, expressed in hours per machine hour (MH), empirical values are taken from the Austrian Federal Forests PLC (Table 4) [28,29]. A distinction is made between delivery on one's own axle and by means of a low-loader. For the transfer of workers, a travel time of one hour (round trip) is assumed for every worker for an eight-hour shift.

**Table 4.** Effort of the relocation of forest machines.

| Machine | Own Axle (h/MH) | Low-Loader (h/MH) | Total (h/MH) |
|---|---|---|---|
| Harvester | 0.035 | 0.048 | 0.083 |
| Forwarder | 0.020 | 0.044 | 0.064 |
| Skidder | 0.008 | 0.042 | 0.050 |
| Tractor | 0.050 | 0.000 | 0.050 |
| Cable Yarder | 0.042 | 0.000 | 0.042 |

Four scenarios have been defined to build the capacity plan: (A) Prefer cable yarder whenever technically possible, (B) Prefer tractor whenever technically possible, (C) Prefer skidder whenever technically possible, (D) Prefer forwarder whenever technically possible.

## 4. Results

### 4.1. Scenario A—Prefer Cable Yarder Whenever Technically Possible

In scenario A, more than 123,000 m$^3$ of wood are harvested on 533 plots with a total area of 197 ha. Including relocation, cable yarders amounting to almost 19,500 h are required. Workers are needed to the extent of more than 64,000 h (Table 5). With an annual utilization of 1500 h, a cable would need almost 13 years to extract the trees. If the harvesting operations needs to be completed within a certain period of time, more machines are needed that work on the harvesting plots at the same time.

**Table 5.** Capacity plan for scenario A.

| | Plots (n) | Area (ha) | Timber Volume (m$^3$) | Machine Hours (h) | Man-Hours (h) | Productivity (m$^3$/h) |
|---|---|---|---|---|---|---|
| No harvesting | 74 | 15.64 | 7341 | - | - | - |
| Chainsaw-Cable yarder | 533 | 196.67 | 123,416 | 18,789 | 56,367 | 6.57 |
| Relocation of cable yarder | - | - | - | 658 | 658 | - |
| Transfer of worker | - | - | - | - | 7046 | |
| Total | 533 | 196.67 | 123,416 | 19,447 | 64,071 | 6.34 |

### 4.2. Scenario B—Prefer Tractor Whenever Technically Possible

In scenario B, cable yarders are required for around 17,100 h and tractors for around 9000 h. Workers are needed to the extent of almost 86,000 h (Table 6). With an annual utilization of 1500 h, a cable yarder would need almost 12 years for timber extracting and a tractor with an annual utilization of 650 h would take around 14 years. If the harvesting operations should be completed within 6 months, 23 cable yarders and 28 tractors are needed to work on the harvesting plots at the same time.

**Table 6.** Capacity plan for scenario B.

| | Plots (n) | Area (ha) | Timber Volume (m$^3$) | Machine Hours (h) | Man-Hours (h) | Productivity (m$^3$/h) |
|---|---|---|---|---|---|---|
| No harvesting | 74 | 15.64 | 7341 | - | - | - |
| Chainsaw-Cable yarder | 408 | 159.34 | 101,814 | 16,535 | 49,605 | 6.16 |
| Relocation of cable yarder | - | - | - | 579 | 579 | - |
| Chainsaw-Tractor | 125 | 37.33 | 21,602 | 8600 | 25,800 | 2.51 |
| Relocation of tractor | - | - | - | 430 | 430 | - |
| Transfer of worker | - | - | - | - | 9426 | - |
| Total | 533 | 196.67 | 123,416 | 26,144 | 85,840 | |

### 4.3. Scenario C—Prefer Skidder Whenever Technically Possible

In scenario C, cable yarders to the extent of around 17,100 h and skidders—depending on the model—of around 2600 to 4800 h are required for the harvesting operation in

windthrown areas. Workers are needed for about 65,000 to 72,000 h (Table 7). To harvest all areas which are technically accessible for a wheeled machine, a skidder would need around 2.5 to 5 years. If all the harvesting operations need to be completed within 6 months, 23 cable yarders and 5 to 10 skidders need to work simultaneously.

**Table 7.** Capacity plan for scenario C.

| | Plots (n) | Area (ha) | Timber Volume (m³) | Machine Hours (h) | Man-Hours (h) | Productivity (m³/h) |
|---|---|---|---|---|---|---|
| No harvesting | 74 | 15.64 | 7341 | - | - | - |
| Chainsaw-Cable yarder | 408 | 158.85 | 101,401 | 16,488 | 49,464 | 6.15 |
| Relocation of cable yarder | - | - | - | 577 | 577 | - |
| Chainsaw-Skidder Timberjack 450 C | 125 | 37.82 | 22,015 | 4578 | 13,734 | 4.81 |
| Relocation of skidder (self) | - | - | - | 37 | 37 | - |
| Relocation of skidder (low-loader) | - | - | - | 192 | 192 | - |
| Transfer of worker | - | - | - | - | 7900 | - |
| Total—Scenario C1 | 533 | 196.67 | 123,416 | 21,872 | 71,904 | |
| Chainsaw-Skidder TAF 690 OP | 125 | 37.82 | 22,015 | 2473 | 7419 | 8.90 |
| Relocation of skidder (self) | - | - | - | 20 | 20 | - |
| Relocation of skidder (low-loader) | - | - | - | 104 | 104 | - |
| Transfer of worker | - | - | - | - | 7110 | - |
| Total—Scenario C2 | 533 | 196.67 | 123,416 | 19,662 | 64,694 | |
| Chainsaw-Skidder TAF 657 | 125 | 37.82 | 22,015 | 3061 | 9183 | 7.19 |
| Relocation of skidder (self) | - | - | - | 24 | 24 | - |
| Relocation of skidder (low-loader) | - | - | - | 129 | 129 | - |
| Transfer of worker | - | - | - | - | 7331 | - |
| Total—Scenario C3 | 533 | 196.67 | 123,416 | 20,279 | 66,708 | |

### 4.4. Scenario D—Prefer Forwarder Whenever Technically Possible

In scenario D, forwarders are used in all areas, if technically possible. As an example, for the case study region Plöcken, 67% of the area was assigned to cable yarders, 32% to forwarders and 1% was not possible to harvest from a technical point of view (Figure 2).

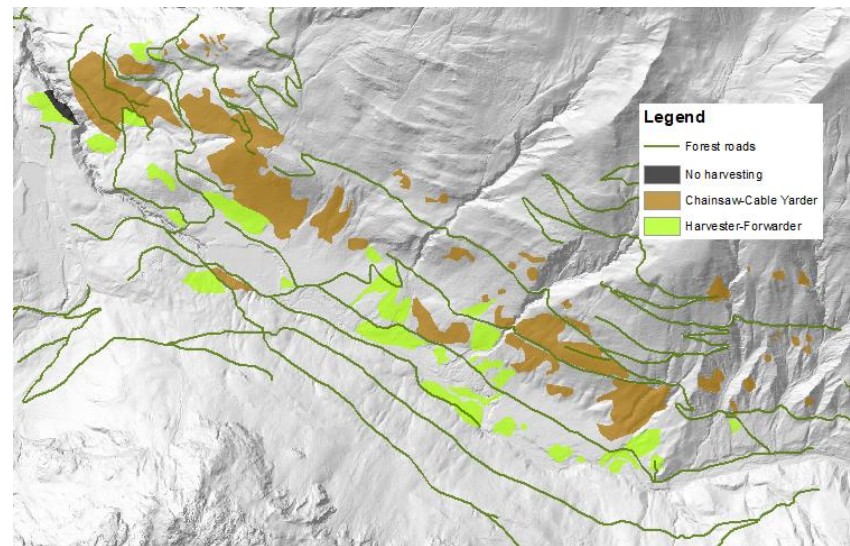

**Figure 2.** Allocation of harvesting areas for chainsaw-cable yarder and harvester-forwarder for the case study region Plöcken.

Including the relocation, cable yarders for around 16,500 h and forwarders for around 900 h are required for the harvesting of all areas. Cable yarders would take almost 12 years

and a forwarder would take around 7 months to harvest all calamity areas together. If the harvesting should be completed within 6 months, 23 cable yarders and more than 1 forwarder are necessary to work on the areas at the same time. It has been shown that scenario D is the most productive one, which means the lowest man-hours were needed for the windthrow processing. Harvesters are required for around 500 to 1000 h, depending on the model for processing the trees in all areas where forwarders operate. Workers are needed for about 58,000 h (Table 8). A harvester is needed for around 6 months to harvest accessible plots in the case study area.

**Table 8.** Capacity plan for harvesters in scenario D.

| | Plots (n) | Area (ha) | Timber Volume (m$^3$) | Machine Hours (h) | Man-Hours (h) | Productivity (m$^3$/h) |
|---|---|---|---|---|---|---|
| No harvesting | 74 | 15.64 | 7341 | - | - | - |
| Chainsaw-Cable yarder | 406 | 158.85 | 101,401 | 16,488 | 49,464 | 6.15 |
| Relocation of cable yarder | - | - | - | 577 | 577 | - |
| Forwarder | 127 | 37.82 | 22,015 | 906 | 906 | 24.31 |
| Relocation of forwarder (self) | - | - | - | 18 | 18 | - |
| Relocation of forwarder (low-loader) | - | - | - | 40 | 40 | - |
| Harvester Kärhä [1] | | same as forwarder | | 921 | 921 | 23.90 |
| Relocation of harvester (self) | - | - | - | 32 | 32 | - |
| Relocation of harvester (Low-loader) | - | - | - | 44 | 44 | - |
| Transfer of worker | - | - | - | - | 6411 | - |
| Total—Scenario D1 | 533 | 196.67 | 123,416 | 19,026 | 58,413 | |
| Model Brzózko [2] | | same as forwarder | | 465 | 465 | 47.37 |
| Relocation (self) | - | - | - | 16 | 16 | - |
| Relocation (Low-loader) | - | - | - | 22 | 22 | - |
| Transfer of worker | - | - | - | - | 6354 | - |
| Total—Scenario D2 | 533 | 196.67 | 123,416 | 18,532 | 57,862 | |
| Model Hagauer [3] | | same as forwarder | | 858 | 858 | 25.67 |
| Relocation (self) | - | - | - | 30 | 30 | - |
| Relocation (Low-loader) | - | - | - | 41 | 41 | - |
| Transfer of worker | - | - | - | - | 6403 | - |
| Total—Scenario D3 | 533 | 196.67 | 123,416 | 18,958 | 58,337 | |

[1] Model Kärhä [2] was developed based on the machines: Ponsse Ergo, John Deere 1270D ECOIII, Logset 8H GT. [2] Model Brzózko [12] was developed based on the machines: Rottne H-14, Valmet 941, Valmet 911. [3] Model Hagauer [11] takes into account the harvesting with the tracked excavator Atlas 1804 LC and the harvester Impex 1650 T.

## 5. Discussion

The present work shows how models on the basis of stand and terrain parameters can be developed to support decision-making for timber harvesting after windthrow events. By including productivity models, it was possible to calculate the time required for harvesting processes and, thus, to estimate the machines and workers required (capacity plan).

The quality of the input data also reflects the quality of the results. For the 564 harvesting plots, the timber volume per hectare, the mean vegetation heights and the DBH of the mean basal area tree were derived from a canopy model and a terrain model. The mean tree volume was calculated using the diameter of the mean basal area tree (proxy for DBH) and the mean vegetation height. The data is approximately normally distributed and does not show any critical outliers. The calculation of the stand densities is also only a rough approximation. There is an area that has densities of over 3000 trees per hectare. This value is to be questioned critically. The digital terrain model does not show any noticeable irregularities. The slope in the study area averages 69%. There are some values that are relatively high, up to 179%. However, these values are not unusual; they indicate the presence of edges of the terrain. The distribution of the extraction distances shows a

noticeable right-skew. However, this can easily be explained by the fact that most areas are well opened-up and only a few areas are difficult to access and therefore have long extraction distances.

The decision as to which productivity model is used for the calculations of the capacity depends on the availability and the quality of the model. Productivity models usually describe local models that have been developed for a specific stand (stand density, tree volume, DBH, etc.) and certain terrain parameters (slope, extraction distance, etc.). The stand and terrain data of the calamity areas in the present study have a strong variability, and therefore it can be that the extreme values of the calamity areas are not covered by the limits of the productivity models used. The harvester models from Kärhä et al. [2] and Hagauer [11] generate realistic values for the study area. The model by Brzózko et al. [12] overestimates productivity and should therefore be questioned critically. Average productivity is twice that of the other two models. Dvořák's model [13] is too specific for the data of the study area. The results have too many outliers and the model was therefore not considered any further. The data generated by the productivity models for tractor and skidder all show realistic values and could, therefore, be used to calculate the time requirements. The forwarder model [19] generates useful values for the study area. The model for the cable yarder [22] shows a few outliers upwards, but for the most part is also well suited (Figure 3). Before developing a capacity plan based on productivity models, it is important to check if the models fit to the local conditions. Furthermore, the availability of different models is probably connected to availability of technologies. This explains why there are a few models for wheeled machines, but only one for the cable yarder.

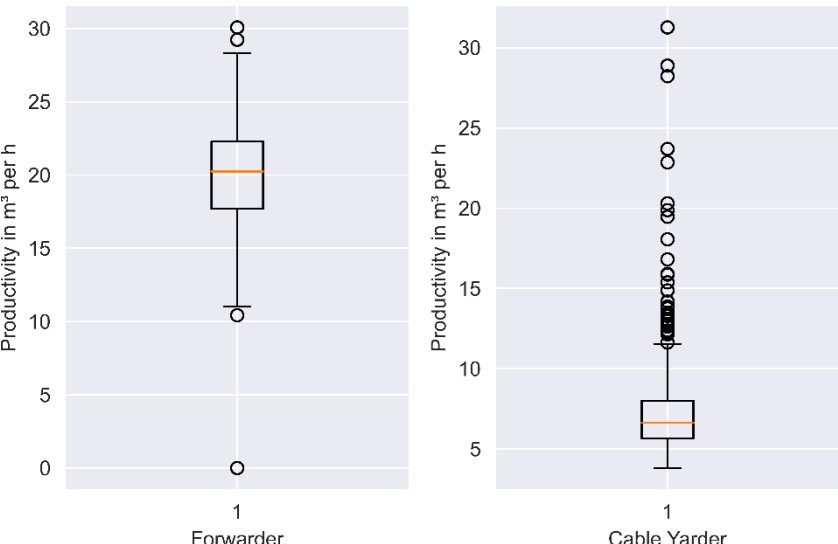

**Figure 3.** Box plot of productivities in the case study area based on the investigated models for forwarders and cable yarders.

## 6. Conclusions

The work provides a basis for decision-making for timber harvesting after windthrow events. Most of the implementation took place on steep terrain. In the future, further investigations should take place in moderately steep or flat terrain to check whether the generated models can be used universally.

For capacity planning, no productivity model for windthrow processing with a chainsaw or processor was found. In the productivity study for extracting with a cable yarder [22], motor-manual processing was carried out, but no time study was available for this process. In the study by Hagauer [11], motor-manual stocking was also carried out, but was also not taken into account in the time study. A productivity study should be carried out for each of these two harvesting systems. Nevertheless, chainsaw operations are usually carried out simultaneously with extraction processes with tractor, skidder and cable yarder, and,

therefore, we assumed that the productivity of the chainsaw operations is the same as the extraction machines.

Finally, it would be advisable to model the chronological sequence of the wood harvesting operations. Suitable approaches are optimization models that take into account the duration of the wood harvesting operations and the relocation of the machines. Other influencing factors would be capacity restrictions of machines or infrastructure, as well as time restrictions (weather). The combination of capacity and sequence plan would map the deployment planning in the windthrow processing in detail. An expansion of the supply chain to include log transport would also be an option here. This is important because the interface between harvest and transport in the mountain forest has to be well coordinated due to the small storage capacities. Optimization and simulation are suitable methods for answering these questions.

**Author Contributions:** Conceptualization, M.K. and K.S.; methodology, M.K., C.G. and A.N.; software, M.K., C.G. and A.N.; validation, C.G., A.N., K.S. and M.L.; formal analysis, K.S.; data curation, M.K. and C.G.; writing—original draft preparation, M.K.; writing—review and editing, C.G., A.N. and K.S.; visualization, M.K. and A.N.; supervision, K.S. and M.L.; project administration, K.S.; funding acquisition, K.S. All authors have read and agreed to the published version of the manuscript.

**Funding:** This research was funded by Federal Ministry of Agriculture, Regions and Tourism, ecoforst GmbH, FHP Kooperationsplattform Forst Holz Papier, Konrad Forsttechnik GmbH, MM Forsttechnik GmbH, grant number 101470.

**Data Availability Statement:** The data presented in this study are available on request from the corresponding author.

**Acknowledgments:** The authors appreciate the support that was given by the forest owners and the team of the Carinthian Forest Service, in particular Clemens Wassermann and Günter Kronawetter. This research was supported by a master thesis at the University of Applied Sciences Technikum Wien supervised by Karl Stampfer, Wolfgang Neussner and Maximilian Lackner.

**Conflicts of Interest:** The authors declare no conflict of interest.

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
