# Peer review of "Capacity Planning of Timber Harvesting in Windthrow Areas"

_forests, doi:10.3390/f13020350_

Round 1
Reviewer 1 Report
Dear Authors,
Congratulations! Good work!
Your paper is very topical and interesting.
Only some small revision before publishing, for instance:
- Lines 37 and 38: Latin names in Italics.
- Table 1. Average piece volume in m3; I prefer log volume. >> Average log volume in m3. // Same in line 128.
- Line 321. ... do not show any outliers. // Is it true?
Thank you!
Author Response
Dear reviewer,
thank you spending your time to read our manuscript and for the valuable comments. We tried to consider all your suggestions which can be seen in the following attachment.
Kind regards
Martin Kühmaier, Christoph Gollob, Arne Nothdurft, Maximilian Lackner, Karl Stampfer

Reviewer 2 Report
Dear authors,
interesting article regarding modelling of salvage logging after wind storms. Only minor improvements are necessary to improve the article.
Sincerely,
reviewer

Author Response

(The authors gave the same response as above.)
